# Peer review of "Manufacture of Carbon Materials with High Nitrogen Content"

_materials, 2022, doi:10.3390/ma15072415_

Round 1

Reviewer 1 Report

Dear Authors

This manuscript is focused on the series of activated carbons derived from MOFs (ZIF-8, ZIF-67) are presented. The 2-methylimidazole that is the linker of the studied MOFs during pyrolysis is transformed into carbon with a high nitrogen content up to 20 % at. by pyrolyzing the MOF. The manuscript presented concerns an interesting and actual subject. The following suggestion and comments should be taken:

  1. The overall English needs to be improved. Please seek guidance from a native English speaker if possible ("the" "a", commas, plural form and others could be corrected).
  2. The introduction section needs enhancement 1-3 sentences. Please add in the introduction - more information about other carbons modified with melamine and applications as electrocatalysts in ORR. Please cite: (1) Materials 202114(9), 2448; https://doi.org/10.3390/ma14092448 (2) Materials 2022, 15(3), 821; https://doi.org/10.3390/ma15030821 (3) Materials 2020, 13(12), 2756; https://doi.org/10.3390/ma13122756
  3. Figure 1a. Please correct this image for better quality.
  4. Table 1. Could authors add Vtotal and [Vmesopore/Vtotal x 100%], that is, the share of mesopores in the total volume? in the table?
  5. Figure 3. Adsorption-desorption isotherms. Please describe the hysteresis loops in the text. What does appear it means?
  6. Could authors add Brunauer-Emmett-Teller (BET) Surface Area Analysis and Pore Size and Volume Analysis of obtained materials? I feel this paper has not given an extensive report on Brunauer-Emmett-Teller (BET) Surface Area Analysis and Barrett-Joyner-Halenda (BJH) Pore Size and Volume Analysis, please enhancement it. The authors should clarify this point. How do these features affect the final product and its properties?
  7. Highly doped materials by different elements. Is good or not? Please explain it.
  8. Could the authors include the standard deviation of all methods?
  9. Please add a new figure with high-resolution X-ray photoelectron spectra for C1s

Author Response

Reviewer 1

The overall English needs to be improved. Please seek guidance from a native English speaker if possible ("the" "a", commas, plural form and others could be corrected).

The manuscript has been rewritten and checked with a native English speaker

The introduction section needs enhancement 1-3 sentences. Please add in the introduction - more information about other carbons modified with melamine and applications as electrocatalysts in ORR.

Please cite: 

(1) Materials 2021, 14(9),2448;https://doi.org/10.3390/ma14092448 

(2) Materials 2022, 15(3),821;https://doi.org/10.3390/ma15030821

(3) Materials 2020, 13(12), 2756; https://doi.org/10.3390/ma13122756

The introduction has been modified following your instructions

Figure 1a. Please correct this image for better quality.

Done

Table 1. Could authors add Vtotal and [Vmesopore/Vtotal x 100%], that is, the share of mesopores in the total volume? in the table?

Done

Figure 3. Adsorption-desorption isotherms. Please describe the hysteresis loops in the text. What does appear it means?

In adsorption it is usual to find this phenomenon where the desorption branch goes above the adsorption branch. In most cases, it is not really in "equilibrium". In our case the hysteresis is very marked, so it is due to that fact. Basically there are two possibilities, one occurs in the so-called slit pore, which is not our case, since it is common in lamellar systems such as montmorillonite, and the other is the so-called ink bottle  pore. A very detailed description can be found in the book by Gregg & Sing (1967). Using the Kelvin equation they show that in the adsorption branch the relevant pore radius is internal to the pore and in desorption the relevant radius is the radius of the pore entrance.

In our case, the carbon material is preferentially microporous, which implies that the pore size is less than 2 nm, while the size of the observed particles is much larger. If we manage to extract the cobalt, the cavities it leaves behind are very large, greater than 2nm. Therefore, the ink bottle  pore described above is generated, for this reason, when more cobalt is eliminated, more cavities are generated, and the hysteresis increases.

Could authors add Brunauer-Emmett-Teller (BET) Surface Area Analysis and Pore Size and Volume Analysis of obtained materials? I feel this paper has not given an extensive report on Brunauer-Emmett-Teller (BET) Surface Area Analysis and Barrett-Joyner-Halenda (BJH) Pore Size and Volume Analysis, please enhancement it. The authors should clarify this point. How do these features affect the final product and its properties?

The authors think that doing extra analysis using BJH is not necessary. That analysis is very popular now, we know, it was originally intended for mesoporous materials, which is not our case. His other indications have been taken in the rewriting of the manuscript

Highly doped materials by different elements. Is good or not? Please explain it.

In this manuscript, what we want to value is that it is possible to prepare carbons with high porosity, and with a high nitrogen content. That of all the strategies developed to date they are not capable of producing it and that the use of MOF opens a new window to achieve it. Some authors had already used it (MOFs), but they had not carried out a study as extensive and detailed as the one we present.

Through text it is shown that for some applications this type of carbon is necessary. We have focused on two CO2 captures, where those obtained from ZIF-8 are more efficient. And in electrochemistry where the most efficient are those obtained from ZIF-67. However, Professor Gascón has already used the carbons obtained from ZIF-67 for catalysis, obtaining exceptional results. There is an application that is under development and not yet reported is the use of magnetic adsorbents as in our case. In literature what we find are core/shell structures where the core is a magnetic material (Fe3O4), and the shell are different adsorbents such as SiO2

Could the authors include the standard deviation of all methods?

Done

Please add a new figure with high-resolution X-ray photoelectron spectra for C1s 

Done

Reviewer 2 Report

The work presents a set of interesting results. However, the presentation itself does not allow to understand the main aims of the research and, hence, the main results. To better understanding the authors should revise the text in order to answer the following questions.
-  What was new in the work, and what was the central purpose of it? The preparation of new material with high content of nitrogen? The investigation of it? 
- What new approaches were used? 
The authors should explain the main points of the investigation to readers.

Author Response

Reviewer 2

The work presents a set of interesting results. However, the presentation itself does not allow to understand the main aims of the research and, hence, the main results. To better understanding the authors should revise the text in order to answer the following questions.
-  What was new in the work, and what was the central purpose of it? The preparation of new material with high content of nitrogen? The investigation of it? 
- What new approaches were used?
The authors should explain the main points of the investigation to readers.

The manuscript has been modified taking into account the comments of all reviewers. I hope that in the new version everything is clearer for the readers

Reviewer 3 Report

The paper is interesting and scientifically important. Authors presented a series of activated carbons derived from MOFs with N doping for CO2 adsorption. They provided results of SEM, TEM, BET, X-ray diffraction (XRD) analysis,CO2 adsorption analysis, thermogravimetric analysis, and microstructure and porous systems. 1.Based on the N2 adsorption-desorption isotherms, the pore size distribution should be also provide. 2.The effect of the N function groups on the CO2 adsorption should disccuss more deeper and more carefully. 3. The results should be compared with the works from open literatures. 4. Conclusion needs to be improved.

Author Response

Reviewer 3

The paper is interesting and scientifically important. Authors presented a series of activated carbons derived from MOFs with N doping for CO2 adsorption. They provided results of SEM, TEM, BET, X-ray diffraction (XRD) analysis,CO2 adsorption analysis, thermogravimetric analysis, and microstructure and porous systems.

1.Based on the N2 adsorption-desorption isotherms, the pore size distribution should be also provide.

It has not been done for all the samples, but it has been done for the most relevant ones.

 2.The effect of the N function groups on the CO2 adsorption should disccuss more deeper and more carefully.

Done

  1. The results should be compared with the works from open literatures.

Done

  1. Conclusion needs to be improved. 

Done

Reviewer 4 Report

In this manuscript, the authors examined potential applications of nitrogen doped carbons obtained from ZIF in CO2 capture, catalysis and electrocatalysis. On the whole, this manuscript is an interest work. However, it requires certain corrections. I suggest the editor consider this manuscript after major revision.

  1. The abstract must be corrected so that it contains summarized knowledge gained in this research. This is very vague, especially certain sentences, like "The 2-methylimidazole that is the linker of the studied MOFs during pyrolysis is 14 transformed into a carbon with a high nitrogen content up to 20% at. By pyrolyzing the MOF." Introduce the abbreviation for MOF if you use it in the abstract.
  2. In Introducion part the sentence "In both cases the presence of N in the structure is presented as essential." is superfluous. Since the authors prepared carbon materials from ZIF please provide more informations about these materials and their previous application in the introduction part.
  3. Correct English and typos through the manuscript. Besides, authors should check references in the text and cite them correctly. Uniform the references list.
  4. In line 73 provide information about used furnace, as well as uniform information about used devices.
  5. Can the authors explain what causes the stability of the MOF?
  6. In lines 118-120 the references are missing.
  7. The authors stated "However, after washing, it is observed that the area drops to 50 m2/g, which indicates that the carbon phase has not yet been produced, or it has been partially produced, and it has simply been a collapse of the structure.", whether a similar phenomenon has occurred earlier in the literature? 

Author Response

Reviewer 4

1 The abstract must be corrected so that it contains summarized knowledge gained in this research. This is very vague, especially certain sentences, like "The 2-methylimidazole that is the linker of the studied MOFs during pyrolysis is 14 transformed into a carbon with a high nitrogen content up to 20% at. By pyrolyzing the MOF." Introduce the abbreviation for MOF if you use it in the abstract

.

Done

2 In Introducion part the sentence "In both cases the presence of N in the structure is presented as essential." is superfluous. Since the authors prepared carbon materials from ZIF please provide more informations about these materials and their previous application in the introduction part.

Done

3  Correct English and typos through the manuscript. Besides, authors should check references in the text and cite them correctly. Uniform the references list.

Done

4  In line 73 provide information about used furnace, as well as uniform information about used devices.

Done

5  Can the authors explain what causes the stability of the MOF?

The theoretical studies performed by Sholl and Coudert shows that the origin of the thermal stability is the strong coordination bond between late transition metal such Co or Zn.

6  In lines 118-120 the references are missing.

Done

7  The authors stated "However, after washing, it is observed that the area drops to 50 m2/g, which indicates that the carbon phase has not yet been produced, or it has been partially produced, and it has simply been a collapse of the structure.", whether a similar phenomenon has occurred earlier in the literature? 

As we have previously commented, MOFs are not stable in acid medium. Thus if the MOFs have not been converted to carbon. When we introduce it in an acid medium, it is destroyed and the only area that is observed is that of the little carbon that has been produced. Note that the isotherm is practically the same as that of the original ZIF, as indicated in the text.

Reviewer 4

1 The abstract must be corrected so that it contains summarized knowledge gained in this research. This is very vague, especially certain sentences, like "The 2-methylimidazole that is the linker of the studied MOFs during pyrolysis is 14 transformed into a carbon with a high nitrogen content up to 20% at. By pyrolyzing the MOF." Introduce the abbreviation for MOF if you use it in the abstract

.

Done

2 In Introducion part the sentence "In both cases the presence of N in the structure is presented as essential." is superfluous. Since the authors prepared carbon materials from ZIF please provide more informations about these materials and their previous application in the introduction part.

Done

3  Correct English and typos through the manuscript. Besides, authors should check references in the text and cite them correctly. Uniform the references list.

Done

4  In line 73 provide information about used furnace, as well as uniform information about used devices.

Done

5  Can the authors explain what causes the stability of the MOF?

The theoretical studies performed by Sholl and Coudert shows that the origin of the thermal stability is the strong coordination bond between late transition metal such Co or Zn.

6  In lines 118-120 the references are missing.

Done

7  The authors stated "However, after washing, it is observed that the area drops to 50 m2/g, which indicates that the carbon phase has not yet been produced, or it has been partially produced, and it has simply been a collapse of the structure.", whether a similar phenomenon has occurred earlier in the literature? 

As we have previously commented, MOFs are not stable in acid medium. Thus if the MOFs have not been converted to carbon. When we introduce it in an acid medium, it is destroyed and the only area that is observed is that of the little carbon that has been produced. Note that the isotherm is practically the same as that of the original ZIF, as indicated in the text.

https://doi.org/10.1021/acs.jpcc.7b12058

https://doi.org/10.1063/1.4904818

https://doi.org/10.1021/acs.jpcc.7b12058

https://doi.org/10.1063/1.4904818

Round 2

Reviewer 1 Report

The authors have addressed all comments and the manuscript can be published as is.

Reviewer 2 Report

On the reviewer's opinion, the authors have answered all questions and have improved the text. The article can be published in present form.